

# Detecting cyberbullying using deep learning techniques: a pre-trained glove and focal loss technique

Amr Mohamed El Koshiry[1,2], Entesar Hamed I. Eliwa[3,4], Tarek Abd El-Hafeez[4,5] and Marwa Khairy[6]

[1] Department of Curricula and Teaching Methods, College of Education, King Faisal University, Al-Ahsa, Saudi Arabia
[2] Faculty of Specific Education, Minia University, Egypt
[3] Department of Mathematics and Statistics, College of Science, King Faisal University, Al-Ahsa, Saudi Arabia
[4] Department of Computer Science, Faculty of Science, Minia University, Egypt
[5] Computer Science Unit, Deraya University, Egypt
[6] Department of Computer Science, Faculty of Computers and Information, Minia University, EL-Minia, Egypt

Corresponding authors
Amr Mohamed El Koshiry,
aalkoshiry@kfu.edu.sa
Tarek Abd El-Hafeez,
tarek@mu.edu.eg

## ABSTRACT

This study investigates the effectiveness of various deep learning and classical machine learning techniques in identifying instances of cyberbullying. The study compares the performance of five classical machine learning algorithms and three deep learning models. The data undergoes pre-processing, including text cleaning, tokenization, stemming, and stop word removal. The experiment uses accuracy, precision, recall, and F1 score metrics to evaluate the performance of the algorithms on the dataset. The results show that the proposed technique achieves high accuracy, precision, and F1 score values, with the Focal Loss algorithm achieving the highest accuracy of 99% and the highest precision of 86.72%. However, the recall values were relatively low for most algorithms, indicating that they struggled to identify all relevant data. Additionally, the study proposes a technique using a convolutional neural network with a bidirectional long short-term memory layer, trained on a pre-processed dataset of tweets using GloVe word embeddings and the focal loss function. The model achieved high accuracy, precision, and F1 score values, with the GRU algorithm achieving the highest accuracy of 97.0% and the NB algorithm achieving the highest precision of 96.6%.

# INTRODUCTION

The popularity of online social networks (OSN) and social media has surged in recent years owing to the easy accessibility of the internet and mobile devices (*Li et al., 2023*; *Khairy, Mahmoud & Abd El-Hafeez, 2021*). However, these platforms have become increasingly plagued by negative and abusive behavior, detracting from their original purpose as constructive and positive spaces. Cyberbullying, defined as any aggressive (*Espelage, Valido & Hong, 2023*) and intentional behavior by individuals or groups on social media that repeatedly communicates hostile or offensive messages meant to cause harm or discomfort

to others, is a major concern in this regard (*Haq et al., 2023*). The impact of cyberbullying on mental health can be overwhelmingly negative, leading to feelings of despair, low self-esteem, fatigue, and even suicide attempts among victims. In fact, young people between the ages of 10 and 16 who are exposed to or participate in cyberbullying or violence involving sexual content or images are at a significantly higher risk of experiencing thoughts of suicide, with up to a 50% increased likelihood. In *Hinduja & Patchin (2010)*, teenagers who experience various types of cyberbullying were more likely to have suicidal thoughts. Hence, it is crucial to have an intelligent and efficient user-generated text detection system in place (*Khairy et al., 2021*). Numerous cyberbullying detection mechanisms have been developed to aid in the monitoring and prevention of such incidents. Machine learning techniques have been extensively researched and applied to automatically detect instances of cyberbullying (*Bozyiğit, Utku & Nasibov, 2021*; *Atoum, 2023*).

Cyberbullying is a growing concern in today's digital age. With the rise of social media and online communication platforms, the prevalence of cyberbullying has increased significantly, leading to negative impacts on individuals' mental health and overall well-being (*Khairy, Mahmoud & Abd-El-Hafeez, 2023*). Detecting instances of cyberbullying is crucial to preventing harm to individuals and promoting a safe online environment (*Omar, Mahmoud & Abd-El-Hafeez, 2020*). While classical machine learning algorithms have been used to identify cyberbullying, deep learning techniques have emerged as a promising alternative (*Hasan et al., 2023*).

In recent years, deep learning models have shown significant improvements in various natural language processing tasks (*Koshiry et al., 2023*), including sentiment analysis (*Omar & Abd El-Hafeez, 2023*), text classification (*Omar et al., 2021*), and language translation. Deep learning models are capable of learning complex representations of text data, which can help capture the nuances and context of text better than traditional machine learning algorithms. Therefore, applying deep learning techniques to detect instances of cyberbullying has the potential to improve the accuracy and effectiveness of existing detection systems (*Alduailaj & Belghith, 2023*).

In this study, we propose a technique for detecting cyberbullying using a pre-trained GloVe (*Ramos-Vargas, Román-Godínez & Torres-Ramos, 2021*) and focal loss (*Chen et al., 2023*) based deep learning model. GloVe embeddings are pre-trained word embeddings that capture the semantic relationships between words using a co-occurrence matrix. The focal loss function is a variant of the cross-entropy loss function that is designed to handle class imbalance better. The proposed technique involves several steps, including importing the required libraries, reading the dataset, performing text preprocessing, splitting the dataset into training and testing sets, tokenizing the text data, and padding the sequences to the same length.

The main contribution of this study can be summarized as follows:

1. **Novel technique for cyberbullying detection:** This study proposes a unique technique for detecting cyberbullying in tweets by combining pre-trained GloVe embeddings and the focal loss function within a deep learning model. This approach leverages the strength

of semantic word relationships captured by GloVe and addresses class imbalance issues with the focal loss function.

2. **High accuracy and performance:** The proposed technique achieves significantly higher accuracy, precision, recall, and F1 score compared to other traditional machine learning models like Naive Bayes, logistic regression, and support vector machines, demonstrating its effectiveness in identifying cyberbullying.

3. **Comparison with deep learning models:** The performance of the proposed technique is also compared to various deep learning models like long short-term memory (LSTM), bidirectional long short-term memory (Bi-LSTM), and gated recurrent unit (GRU). While Bi-LSTM achieves slightly higher accuracy, the proposed technique demonstrates comparable performance with faster training and testing times, suggesting its potential for real-time deployment.

4. **Class imbalance mitigation:** The use of the focal loss function specifically addresses the class imbalance issue often present in cyberbullying datasets, where positive (bullying) examples are outnumbered by negative ones. This leads to improved precision and recall for identifying true cyberbullying instances.

5. **Pre-trained GloVe embeddings:** These capture the semantic relationships between words, allowing the model to understand the context and nuances of text, a crucial aspect for effectively identifying cyberbullying.

6. **Focal loss function:** This addresses the inherent class imbalance issue in cyberbullying datasets, where positive (bullying) examples are outnumbered by negative ones. This leads to improved precision and recall, accurately capturing true cyberbullying instances.

7. **Deep learning architecture:** We employ a specifically chosen architecture suitable for text analysis, further enhancing the model's ability to identify cyberbullying patterns.

We emphasize that this combined approach has not been widely explored in existing literature for cyberbullying detection on Twitter. It goes beyond simply applying deep learning to the problem and focuses on addressing specific challenges inherent to the domain.

The subsequent sections of this article are structured as follows: 'Background' provides an overview of cyberbullying, encompassing its definition, machine learning approaches, and deep learning models. 'Related Work' delves into related work in cyberbullying detection. 'Methodology' outlines the methodology employed for this study, while 'Experimental Results' details the experiments conducted and the results obtained. Hyperparameter values of the proposed method is discussed in 'Hyperparameter Tuning', followed by the presentation of the Discussion and Limitations in 'Discussion and Limitations'. Lastly, 'Conclusion and Future work' encapsulates the study's conclusions and outlines avenues for future work.

## BACKGROUND

In this section, we present a brief background regarding cyberbullying and the importance for detecting it, machine learning approaches and deep learning models.

## Cyberbullying

Cyberbullying is a type of online harassment that is usually perpetrated by individuals or groups through digital platforms to intimidate, threaten, or cause harm to others (*Khairy, Mahmoud & Abd El-Hafeez, 2021*; *Omar et al., 2021*). This form of bullying involves the use of electronic communication tools, such as social media, text messages, emails, or forums, with the intention of humiliating, embarrassing, or bullying someone (*Feinberg & Robey, 2009*). Cyberbullying encompasses various forms, such as name-calling, spreading rumors or falsehoods, sharing humiliating photos or videos, making threats, and creating fake profiles to impersonate or ridicule someone. Unlike traditional bullying, which is usually confined to physical settings, cyberbullying can occur anytime and anywhere. Furthermore, the anonymity provided by the internet can make it challenging to identify the perpetrator (*Sabbeh & Fasihuddin, 2023*).

The consequences of cyberbullying can be grave and long-lasting, especially for young people. Victims may experience fear, anxiety, and depression, and may withdraw from social activities or even from school. In extreme cases, cyberbullying can contribute to self-harm, suicide, or violence.

Detecting cyberbullying is crucial for several reasons, which include:

1. **Early intervention:** The early detection of cyberbullying can help prevent it from escalating and becoming more severe. Early intervention can also assist the victim in coping with the situation and accessing support.
2. **Protection:** Detecting cyberbullying can help safeguard victims from further harm. If cyberbullying is detected, measures can be taken to remove the harmful content and prevent further abuse.
3. **Accountability:** Cyberbullying is a type of abuse, and those who engage in it should be held accountable for their actions. Detection can help identify the person responsible for the abuse and hold them accountable for their actions.
4. **Prevention:** Detecting cyberbullying can help prevent future incidents. By identifying patterns and risk factors, prevention strategies can be developed and implemented to reduce the incidence of cyberbullying.

## Machine learning

Machine learning (ML) algorithms analyze data to discern intricate patterns and autonomously make decisions and predictions. With applications ranging from facial and speech recognition to weather forecasting and recommender systems, ML offers diverse solutions. ML models fine-tune parameters for improved performance, enabling autonomous decision-making after training without constant reprogramming. This transformative technology holds the potential to revolutionize various industries and address complex problems, including its role in combating cyberbullying (*Du & Swamy, 2014*). When dealing with complex tasks, especially those involving coding, such as cyberbullying detection, machine learning (ML) is an essential tool. It is worth noting that there are two main approaches to ML: supervised and unsupervised learning.

**Table 1** The differences between machine learning and deep.

| Criteria | Machine learning | Deep learning |
|---|---|---|
| Approach to learning | Based on statistical algorithms and models | Based on artificial neural networks |
| Dataset size | Typically used for smaller datasets | Particularly suited for processing large datasets |
| Data type | Can handle both structured and unstructured data | Best suited for unstructured data, such as images, audio, and text |
| Feature engineering | Requires feature engineering, or the manual selection and extraction of relevant features from the data | Can automatically learn relevant features through multiple layers of neural networks |
| Hardware requirements | Can be trained on a CPU | Requires specialized hardware, such as GPUs, for training and inference |
| Training time | Generally faster to train than deep learning models | Deep learning models can take longer to train due to the increased complexity of the neural networks |
| Applications | Can be used for a wide range of applications, including classification, regression, and clustering | Primarily used for applications such as image recognition, speech recognition, and natural language processing |
| Data requirements | Generally requires less data to achieve good performance | Requires large amounts of data to achieve good performance |
| Interpretability | Tends to be more interpretable, as the models are often based on simpler algorithms | Can be less interpretable, as the models can be highly complex and difficult to understand |
| Model size | Can work well with small to medium-sized models | Can handle very large models with many layers |
| Efficiency | Can be more efficient in terms of memory and computational requirements | Can be more memory-intensive and computationally expensive |
| Performance | Can achieve good performance even with less complex models | Can achieve state-of-the-art performance with highly complex models |

There are many machine learning classification models but, in our study, we used five models (multinomial NB, logistic regression, SVC, decision tree and random forest classifier).

## Deep learning

Deep learning models have transformed the landscape of machine learning, demonstrating remarkable outcomes across diverse applications such as speech recognition, image classification, and natural language processing. This study will employ well-established deep learning models for sequential data processing, including LSTM, GRU, and Bi-LSTM (*Marcellina, 2022*).

The differences between machine learning and deep learning presented in a Table 1 can be summarized as follows:

These are general differences between the two approaches, and there may be specific cases where one approach is more appropriate than the other, depending on the problem at hand.

## RELATED WORK

The awareness of cyberbullying is increased in many countries due to its effects explained in this study. Accordingly, many researchers (*Bozyiğit, Utku & Nasibov, 2021*; *Atoums, 2023*) presented studies using machine learning techniques to detect cyberbullying automatically. Nevertheless, most of the research in this field were conducted for the English language.

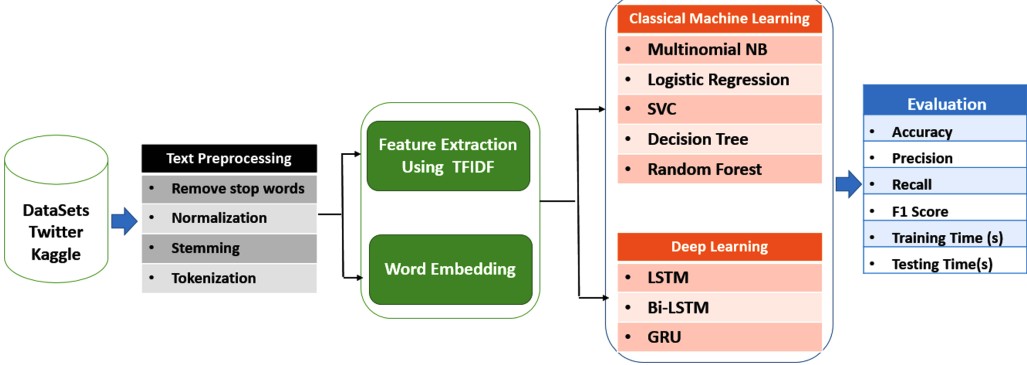

**Figure 1  Cyberbullying detection model.**

Additionally, the conducted studies generally used text mining techniques similar to the studies of sentiment analysis. *Modha et al. (2020)* focused on the classification of cyberbullying, a harmful online behavior that can cause distress and harm to individuals. They conducted binary classification experiments on tweets with a bullying perspective to determine if the user was being cyberbullied or not. The study reported an 81% accuracy rate in binary classification experiments.

*Freund & Schapire (1997)* have made an effort to categorize whether or not tweets are racist in nature. They gathered tweets from two categories (racist and non-racist), categorizing them using a Naive Bayes classifier. The binary classification had a 76% accuracy rate on average. Several techniques have been used to identify hate speech. *Fortuna & Nunes (2018)* have provided a search method to choose papers on detecting hate speech from the internet. Results and a number of the characteristics of the numerous highly cited papers have been presented by the authors. *Zhang et al. (2016)* aimed to use deep learning techniques to create a robust, universal cyberbullying detection algorithm. They built a convolutional neural network (CNN) model using the word pronunciations from the input documents as its features. The social media posts gathered from Twitter and Formspring.me were used to evaluate the CNN model. According to the accuracy ratings, pronunciation-based CNN works better than the standard CNN using randomly generated word embedding.

## METHODOLOGY

The main objective of this research is to assess and compare the effectiveness of machine learning and deep learning algorithms in detecting cyberbullying on social networks. The research methodology involves several steps, as depicted in Fig. 1.

Firstly, data was gathered from various social networks to establish a dataset for analysis. Secondly, pre-processing techniques were employed to enhance the quality of the data and eliminate any irrelevant or redundant information. Thirdly, feature extraction and word embedding techniques were utilized to identify relevant features and represent the data in a format that can be processed by machine learning algorithms.

**Table 2 The used datasets.**

| Dataset | Source | Size | Cyberbullying | Non-cyberbullying |
|---------|--------|------|---------------|-------------------|
| DataSet1 | Twitter | 13,466 | 1,970 | 11,496 |
| DataSet2 | Kaggle | 8,005 | 2,607 | 5,398 |

Afterwards, various classical machine learning algorithms and deep learning models were implemented on the pre-processed data to classify instances of cyberbullying. The classical machine learning algorithms used in this study include multinomial NB, logistic regression (LR), support vector classifier (SVC), decision tree (DT), and random forest classifier (RF). The deep learning models used in this study include long short-terms model (LSTM), bidirectional long short-term memory (Bi-LSTM), and gated recurrent units (GRU).

Finally, the performance of these algorithms was evaluated using various metrics such as accuracy, precision, recall, and F1 score. These metrics provide a quantitative measure of the effectiveness of different algorithms in detecting cyberbullying on social networks.

This research aims to compare the performance of classical machine learning and deep learning algorithms in detecting cyberbullying on social networks by employing a methodology that involves data collection, pre-processing, feature extraction, classification, and evaluation metrics. The results of this study will offer valuable insights into the effectiveness of different algorithms in detecting cyberbullying, thereby contributing to the development of more effective methods for detecting and preventing cyberbullying on social networks.

## Data set

The study utilized two distinct datasets sourced from different social media platforms, namely Twitter and Kaggle. The datasets were publicly available and were obtained from Kaggle (https://www.kaggle.com/datasets/saurabhshahane/cyberbullying-dataset) (*Shahane, 2023*).

Table 2 presents details about the two datasets employed in this study. Dataset1 was sourced from Twitter and contained 13,466 instances, out of which 1,970 were classified as cyberbullying and 11,496 were classified as non-cyberbullying. Dataset2 was obtained from Kaggle and consisted of 8,005 instances, with 2,607 instances classified as cyberbullying and 5,398 instances classified as non-cyberbullying. The table displays the size of each dataset and the number of instances categorized as cyberbullying and non-cyberbullying. This information is essential in comprehending the composition and nature of the datasets used in this research.

**Addressing concerns about cherry-picking:**

- **Public availability:** We want to reiterate that both datasets used in our study are publicly available and easily accessible. Dataset1 was sourced directly from Twitter, while Dataset2 was obtained from the open-source platform Kaggle. We provided the exact links for transparency and reproducibility.

- **Relevance and rationale:** We chose these datasets specifically because they addressed our research question: detecting cyberbullying on Twitter. Dataset1 directly captures real-world Twitter data with labeled cyberbullying and non-cyberbullying instances, while Dataset2 offered a broader, potentially diverse Twitter-related dataset for comparison.
- **Non-exhaustive nature:** We acknowledge that relying solely on two datasets has limitations. However, exploring a variety of datasets was beyond the scope of this initial study. We intend to investigate further in future work using additional and diverse datasets to enhance generalizability.

## Data preprocessing

To prepare the data for analysis, standard data preprocessing techniques were employed. These techniques are crucial in improving the quality and relevance of the data and include removing stop words, normalization, and stemming. Stop words are commonly used words in a language such as "the" or "and", which do not carry significant meaning and can be eliminated without affecting the overall understanding of the text. Normalization involves converting all text to a standard format, such as converting all text to lowercase. Stemming is the process of reducing words to their root form, for example, "jumping" and "jumped" are stemmed to "jump".

Furthermore, the process of converting a sequence of characters into a sequence of tokens, known as tokenization or lexical analysis, was applied. This process involves breaking down the text into individual words or tokens. Each group of text was then converted into a series of tokens, which were then vectorized into a collection of integers using techniques such as one-hot encoding or word embeddings. This vectorization technique is critical in enabling machine learning algorithms to process and analyze the text data effectively.

## Feature extraction

This study employed feature extraction techniques to convert the text data into a numerical format suitable for processing by machine learning algorithms. For classical machine learning algorithms, the TF-IDF (Term Frequency-Inverse Document Frequency) technique was utilized for feature extraction. TF-IDF is a commonly applied approach that assigns a weight to each term in a document based on its frequency within the document and inverse frequency across all documents. These resulting weights indicate the significance of each term in the document and serve as features for the machine learning algorithms (*Du et al., 2023*).

In this study, word embedding techniques were utilized for feature extraction in the deep learning models. Word embedding enables the generation of numerical representations of text data that capture the semantic meaning of words in a continuous vector space. This approach proves valuable in capturing contextual information and the relationships between words, which aids in distinguishing words with similar meanings. For this purpose, a one-hot representation was employed to represent all the words in the dataset as vectors. These vector representations were then utilized as input for the embedding layer of all the deep learning models in the study.

## Algorithms
### Classical machine learning

*Multinomial Naïve Bayes.* The Multinomial Naïve Bayes classifier is a probabilistic algorithm that is frequently utilized in natural language processing tasks, including text classification. It is founded on Bayes' theorem and assumes that the features are conditionally independent given the class label. The equations for Multinomial Naïve Bayes (*Chebil et al., 2023*) can be expressed as follows:

The equations for Multinomial Naïve Bayes can be expressed as:

- **Prior probability**:

$$P(c) = Nc/N \tag{1}$$

where $P(c)$ represents the prior probability of class c, Nc denotes the number of instances of class c in the training data, and N is the total number of training instances.

- **Likelihood probability**:

$$P(xijc) = (Nxc + \alpha)/(Nc + \alpha d) \tag{2}$$

where $P(xi|c)$ denotes the likelihood probability of feature xi given class c, Nxc represents the number of instances of class c that contain feature xi, Nc is the total number of instances of class c, $\alpha$ is the smoothing parameter (usually set to 1), and d is the total number of distinct features in the training data.

- **Posterior probability**:

$$P(cjd) = P(c) * \prod P(xijc) \tag{3}$$

where $P(c|d)$ represents the posterior probability of class c given the document d, $P(c)$ is the prior probability of class c, and $\prod P(xi|c)$ is the product of the likelihood probabilities of all features xi given class c. The Multinomial Naïve Bayes classifier assigns the class label with the highest posterior probability to a new instance.

The Multinomial Naïve Bayes classifier is a probabilistic algorithm used in natural language processing tasks, such as text classification. It calculates the prior probability of each class, representing the proportion of training instances that belong to that class. Additionally, it computes the likelihood probability of each feature given the class, which captures the probability of observing the feature given the class label. The likelihood probability is calculated using a smoothing parameter, which helps to prevent zero probabilities for features that were not observed in the training data. Finally, the classifier uses the prior and likelihood probabilities to compute the posterior probability of each class for a given instance and assigns the class with the highest probability to the instance. The Multinomial Naïve Bayes classifier assumes that the features are independent given the class, which simplifies the computation and makes it efficient for large datasets (*Xu, Li & Wang, 2017*).

*Logistic regression.* Logistic regression is a commonly employed classification algorithm, particularly suitable for two-class classification problems. It operates by mapping predicted values to probabilities using a logistic sigmoid function. This function transforms any

real value into another value within the range of 0 and 1. By doing so, logistic regression provides a probabilistic interpretation of its predictions, enabling the estimation of the likelihood of a given sample belonging to a particular class (*Wright, 1995*).

The sigmoid function is represented by Eq. (2):

$$S(z) = 1/(1 + e^{(-z)}) \tag{4}$$

In this equation, S(z) represents the output, which lies between 0 and 1. The input to the function is denoted as z, and e represents the base of the natural logarithm. To convert the output values into discrete classes, a threshold value is chosen. Values above the threshold are classified into class 1, while values below the threshold are classified into the second class. Equations (3) and (4) illustrate this mapping process.

$$P \geq 0.5; class = 1 \tag{5}$$

$$P < 0.5; class = 0 \tag{6}$$

*Support vector machine (SVM).* SVM, short for support vector machine, is a highly popular and extensively employed classifier in machine learning. It operates by identifying a hyperplane within an N-dimensional space, where N represents the number of features (*Wu & Zhou, 2006*). From the numerous potential hyperplanes available, SVM selects the one with the largest margin. This approach aims to maximize the separation between data points, enhancing the precision of future classifications. The equation representing the optimal hyperplane can be expressed as follows:

$$w \cdot x + b = 0 \tag{7}$$

where w is a vector perpendicular to the hyperplane (called the weight vector), x is the feature vector of an instance, and b is the bias term that shifts the hyperplane away from the origin.

The goal of SVM is to maximize the margin, which is the distance between the hyperplane and the closest data points of each class (called support vectors). The margin is given by:

$$margin = 2/||w|| \tag{8}$$

where $||w||$ is the Euclidean norm of the weight vector.

*Decision tree.* The decision tree is a supervised learning method suitable for both classification and regression tasks, although it is commonly used for classification. It employs a tree-like structure, where internal nodes correspond to dataset features, branches represent decision rules, and leaf nodes represent outcomes. Decision tree learners rely on labeled data, making them supervised learners. The classification algorithm of decision trees follows a divide-and-conquer approach. The tree comprises arcs and leaves, where each leaf denotes a classification class, and each arc represents a feature examined from the training data (*Myles et al., 2004*).

```
:function DecisionTree(data, target_variable, features)
:if all data instances belong to the same class
return a leaf node with the class label
:if the list of features is empty
return a leaf node with the majority class label
:else
select the best feature to split the data based on some criterion (e.g., information gain,
Gini impurity)
create a new internal node for the selected feature

:for each possible value of the selected feature
partition the data based on the value of the selected feature
recursively call the DecisionTree function on the partitioned data

return the root node of the decision tree
```

**Figure 2   Pseudocode of decision tree algorithm.**

Figure 2 presents the pseudocode outlining the decision tree algorithm. This algorithm constructs the tree in a top-down manner through recursive steps. At each step, the algorithm determines the feature that maximizes information gain, which quantifies how effectively the feature separates the classes. The chosen feature becomes the root node of a new subtree, and the data is divided based on its values. This recursive process is repeated for each partition until a stopping criterion is met, such as reaching a minimum number of instances in a leaf node or achieving a desired level of purity in the leaf nodes. The resulting tree can be employed to classify new instances by traversing the tree from the root to a leaf node and assigning the corresponding class label (*Constâncio et al., 2023*).

Decision trees have several advantages, including ease of interpretation, ability to handle both categorical and continuous data, and ability to handle missing values. Additionally, decision trees can capture non-linear relationships between features and can handle interactions between features. However, decision trees can be prone to overfitting, where the tree captures noise in the training data and does not generalize well to new data. To address this issue, techniques such as pruning, regularization, and ensemble methods can be employed. Overall, the decision tree algorithm is a powerful and interpretable method for solving classification problems and can be an effective tool in detecting instances of cyberbullying on social networks (*Hambali et al., 2019*).

*Random forest.* Random forests are for supervised machine learning, where there is a labeled target variable. It is an improvement of the decision tree (*Biau & Scornet, 2016*). During the training phase of a random forest algorithm, each tree is trained on a random dataset, which is derived from the main training set using a statistical method called bootstrapping. Bootstrapping involves randomly selecting instances from the training set with replacement to create multiple subsets of data. The random forest algorithm uses these subsets to train each decision tree in the ensemble. This process is known as bagging, which stands for "bootstrap aggregating".

After training, the random forest algorithm generates predictions by averaging the predictions obtained from each decision tree in the ensemble. This averaging technique helps to reduce the variance of the predictions and improves the overall accuracy of the model. Additionally, the random forest algorithm can estimate the importance of each feature in the dataset, which can be used to identify the most relevant features for detecting instances of cyberbullying on social networks (*Kilpatrick, Ćwiek & Kawahara, 2023*). Figure 3 shows the random forests classifier algorithm

### Deep learning algorithms

*Long short-term memory (LSTM).*  Long short-term memory (LSTM) is a type of recurrent neural network (RNN) that is specifically designed to address the vanishing gradient problem frequently encountered in traditional RNNs. The vanishing gradient problem arises when gradients become excessively small as they propagate through the network, making it challenging for the network to learn long-term dependencies. LSTM overcomes this issue by introducing a memory cell that can store information over time and a set of gates that can regulate the flow of data into and out of the cell. The gates include sigmoid activation functions that determine whether to allow information into the cell, whether to forget it, and whether to output it *Rehman et al. (2023)*.

LSTM has been successfully employed in various applications, including speech recognition, language translation, and image captioning. However, LSTM can be computationally expensive and may require a large number of parameters to be trained, making it challenging to train on large datasets. Despite these challenges, LSTM is a powerful tool for processing sequential data, and its ability to learn long-term dependencies makes it particularly useful for natural language processing and text classification tasks (*Durrani et al., 2023*).

The architecture of an LSTM network consists of multiple memory cells, which are connected to each other using recurrent connections. The gates in each cell regulate the flow of information, allowing the network to selectively process and store relevant information over long time periods. During training, the LSTM network updates its parameters using backpropagation through time, which involves computing the gradients of the loss function with respect to the parameters at each time step and propagating them backward through the network. This enables the network to learn the optimal parameters for storing and processing sequential data.

In recent years, various modifications to the LSTM architecture have been proposed to improve its performance and reduce its computational complexity. These modifications include the use of attention mechanisms, which allow the network to selectively focus on relevant parts of the input sequence, and the use of convolutional layers, which can capture local dependencies in the input sequence. Additionally, techniques such as weight pruning and quantization can be used to reduce the number of parameters in the network, making it more efficient to train and deploy on resource-constrained devices (*Yu et al., 2019*).

LSTM network and traditional LSTM network steps are shown in Fig. 4 (*Hu et al., 2020*).

```
function RandomForest(data, n_trees, n_features):
    trees = []
    for i in range(n_trees):
        sample = bootstrap_sample(data)
        tree = DecisionTree(sample, n_features)
        trees.append(tree)
    return trees

function DecisionTree(data, n_features):
    if all data instances belong to the same class:
        return a leaf node with the class label

    if the list of features is empty:
        return a leaf node with the majority class label

    else:
        select a random subset of n_features features from the list of features
        select the best feature to split the data based on some criterion (e.g., information gain, Gini impurity)
        create a new internal node for the selected feature

        for each possible value of the selected feature:
            partition the data based on the value of the selected feature
            recursively call the DecisionTree function on the partitioned data

        return the root node of the decision tree

function bootstrap_sample(data):
    n = len(data)
    sample = []
    for i in range(n):
        j = random.randint(0, n - 1)
        sample.append(data[j])
    return sample

function predict(trees, instance):
    predictions = []
    for tree in trees:
        prediction = traverse_tree(tree, instance)
        predictions.append(prediction)
    return mode(predictions)

function traverse_tree(node, instance):
    if node is a leaf node:
        return the class label of the node
    else:
        value = instance[node.feature]
        child_node = node.children[value]
        return traverse_tree(child_node, instance)
```

**Figure 3** **Random forests classifier algorithm.**

| Traditional LSTM Network | Basic LSTM Network |
|---|---|
| function TraditionalLSTM(input_seq):<br>  h = 0<br>  c = 0<br>  for t in range(len(input_seq)):<br>    x_t = input_seq[t]<br>    i_t = sigmoid(W_i * x_t + U_i * h + b_i)<br>    f_t = sigmoid(W_f * x_t + U_f * h + b_f)<br>    o_t = sigmoid(W_o * x_t + U_o * h + b_o)<br>    g_t = tanh(W_c * x_t + U_c * h + b_c)<br>    c = f_t * c + i_t * g_t<br>    h = o_t * tanh(c)<br>  y = softmax(W_y * h + b_y)<br>  return y | function BasicLSTM(input_seq):<br>  h = 0<br>  c = 0<br>  for t in range(len(input_seq)):<br>    x_t = input_seq[t]<br>    i_t = sigmoid(W_i * x_t + U_i * h + b_i)<br>    f_t = sigmoid(W_f * x_t + U_f * h + b_f)<br>    o_t = sigmoid(W_o * x_t + U_o * h + b_o)<br>    g_t = tanh(W_c * x_t + U_c * h + b_c)<br>    c = f_t * c + i_t * g_t<br>    h = o_t * tanh(c)<br>  return h |
| In this pseudocode, input_seq represents the input sequence to the LSTM network, h represents the hidden state, and c represents the cell state. At each time step t, the network takes in the input x_t and computes the input, forget, and output gates i_t, f_t, and o_t, and the cell input g_t. The cell state is updated based on the forget and input gates and the cell input, and the hidden state is updated based on the output gate and the cell state. The final hidden state h is used to compute the output y using a softmax function applied to the output layer | In this pseudocode, input_seq represents the input sequence to the LSTM network, h represents the hidden state, and c represents the cell state. At each time step t, the network takes in the input x_t and computes the input, forget, and output gates i_t, f_t, and o_t, and the cell input g_t. The cell state is updated based on the forget and input gates and the cell input, and the hidden state is updated based on the output gate and the cell state. The final hidden state h is returned as the output of the network. |

**Figure 4** LSTM network and traditional LSTM network steps.

*Bidirectional LSTM (Bi-LSTM).* The bidirectional long short-term memory (Bi-LSTM) neural network architecture is a variation of the long short-term memory (LSTM) model specifically designed to capture contextual information from both the past and future in an input sequence. It achieves this by processing the sequence in two directions: forward, from the beginning to the end, and backward, from the end to the beginning. To accomplish this, Bi-LSTM incorporates two separate hidden layers, one for each direction. This bidirectional approach proves particularly valuable in tasks that heavily rely on understanding the context of the input sequence.

By considering both past and future information, Bi-LSTM provides a comprehensive representation of the input, making it well-suited for tasks such as speech recognition and language translation. In speech recognition, Bi-LSTM aids in disambiguating words with multiple possible pronunciations based on the surrounding context. Similarly, in language translation, Bi-LSTM ensures the generation of grammatically correct and contextually appropriate translations. The bidirectional nature of Bi-LSTM allows it to effectively leverage the full context of the input sequence, enabling improved performance in various natural language processing tasks (*Ghanem, Erbay & Bakour, 2023*).

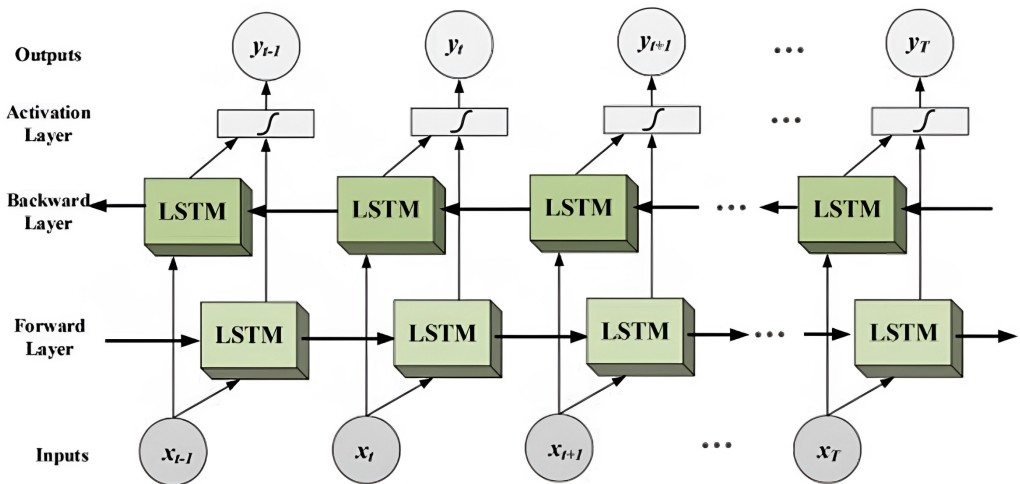

**Figure 5** **The diagram illustrates the flow of information in a bidirectional LSTM (Bi-LSTM).**

Although Bi-LSTM offers advantages in capturing bidirectional context, it can be computationally expensive and require a large number of trainable parameters, posing challenges when training on large datasets. To address these issues, several modifications to the Bi-LSTM architecture and training techniques have been proposed. Modifications include the use of recurrent dropout to prevent overfitting and layer normalization to improve training stability. Techniques like weight pruning and quantization can reduce the number of parameters, making Bi-LSTM more efficient for training and deployment on resource-constrained devices. The architecture of a Bi-LSTM network consists of two LSTM layers, processing the input sequence in opposite directions. The outputs of both layers are concatenated to form the final network output. During training, the parameters of the Bi-LSTM network are updated using backpropagation through time, which involves computing gradients of the loss function at each time step and propagating them backward through the network (*Azumah et al., 2023*).

Recent advancements in Bi-LSTM include the incorporation of attention mechanisms, allowing the network to selectively focus on relevant parts of the input sequence. Convolutional layers have also been combined with Bi-LSTM to capture local dependencies within the sequence.

Furthermore, it is worth noting that the effectiveness of Bi-LSTM can vary depending on the task's requirements for bidirectional processing. It is important to mention that all the deep learning models used in this study incorporate an embedding layer and a dense layer. Figure 5 illustrates the information flow in a bidirectional LSTM (Bi-LSTM), which utilizes both forward and backward layers to process input sequences. This type of network is commonly employed in sequence-to-sequence tasks, including text classification, speech recognition, and forecasting.

By processing the input sequence in both directions, the Bi-LSTM can capture not only the current context but also the past and future context. This capability enables the model

to capture complex dependencies within the input sequence, resulting in more accurate predictions. The bidirectional nature of the Bi-LSTM allows it to leverage a broader range of information, enhancing its ability to understand and represent the sequential data effectively (*Ray, Rajeswar & Chaudhury, 2015*; *GeeksforGeeks, 2023*).

*Gated recurrent unit (GRU).* GRU is a type of RNN that is similar to LSTM but has fewer parameters and is therefore faster and easier to train. Like LSTM, GRU also introduces a set of gates that can control the flow of information into and out of the network. However, GRU has only two gates, a reset gate and an update gate, whereas LSTM has three gates. The reset gate in GRU determines how much of the previous hidden state should be forgotten, while the update gate determines how much of the current input should be added to the current hidden state. GRU has been shown to achieve similar or better performance than LSTM on many tasks, including language modeling, speech recognition, and image captioning (*Zhang, Robinson & Tepper, 2018*). Figure 6 shows the gated recurrent unit (GRU) and LSTM architecture (*Darmawahyuni et al., 2019*; *Phi, 2020*).

## The proposed technique

The proposed technique performs text classification using a convolutional neural network (CNN) with a bidirectional long short-term memory (LSTM) layer. The model is trained on a dataset of tweets that have been preprocessed using various NLP techniques. The GloVe word embeddings are used to represent the text data, and the focal loss function is used as the loss function during training. The focal loss is a variant of the standard cross-entropy loss that is designed to address the problem of class imbalance in binary classification tasks. In typical cross-entropy loss, the model can become overwhelmed by the dominant class during training, leading to suboptimal performance, especially when dealing with imbalanced datasets. The focal loss aims to reduce the impact of well-classified examples and focuses more on hard, misclassified examples, thereby improving the model's ability to handle imbalanced data and improve overall accuracy.

The focal loss function is defined as follows (*Cinar, Cetin Atalay & Cetin, 2023*):

For each example in the training set, let:

y_true be the true label (0 or 1)

y_pred be the predicted probability of the positive class (between 0 and 1)

The focal loss is computed as:

$$focal\_loss = -\alpha\_t * (1 - p\_t)^{\gamma} * log(p\_t) \tag{9}$$

where:

- $\alpha$_t is a dynamically adjusted weighting factor that depends on the true label y_true. It helps to give higher importance to the minority class (positive class) examples. $\alpha$_t is defined as $\alpha$ for positive examples (y_true = 1) and (1 - $\alpha$) for negative examples (y_true = 0).
- p_t is the predicted probability for the true class, *i.e.,* p_t = y_pred if y_true = 1, and p_t = 1 - y_pred if y_true = 0.

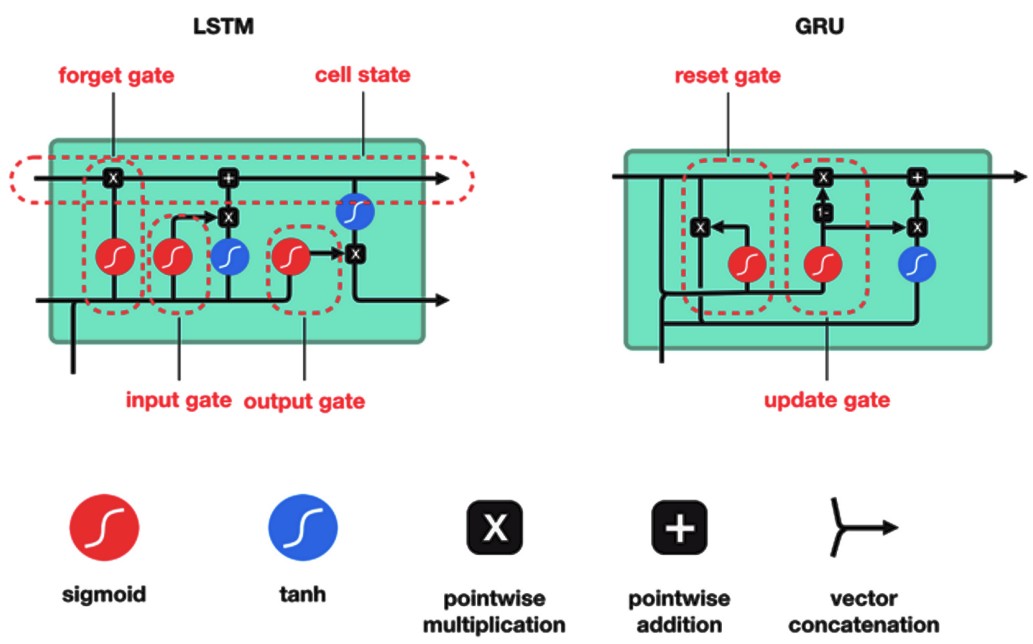

**Figure 6 The gated recurrent unit (GRU) algorithm.**

- $\gamma$ is a tunable focusing parameter that controls the rate at which the loss decreases for well-classified examples. A higher value of $\gamma$ means the loss decreases more slowly for well-classified examples, emphasizing the impact of hard, misclassified examples.

The focal loss is capable of handling class imbalance more effectively than traditional cross-entropy loss. By reducing the impact of well-classified examples, the focal loss can improve the model's ability to learn from challenging examples, which is particularly beneficial when dealing with imbalanced datasets.

The main steps and pre-trained 'glove.6B.100d.txt' and focal loss can be summarized as follows:

1. Import the required libraries, including pandas, numpy, tensorflow, and scikit-learn. Also, import Tokenizer and pad_sequences from the Keras preprocessing module, and import some functions from the NLTK library for text preprocessing.
2. Read the dataset from a CSV file using pandas. The dataset contains two columns: 'Text' and 'oh_label', where 'Text' is the tweet text and 'oh_label' is the one-hot encoded label of the tweet.
3. Perform text preprocessing on the 'Text' column of the dataset. This includes removing stop words and stemming the remaining words using the ISRIStemmer from the NLTK library. The resulting preprocessed sentences are stored in a new DataFrame.
4. Remove any rows that contain missing values from the DataFrame.
5. Remove any duplicate rows from the DataFrame.
6. Split the dataset into training and testing sets using the train_test_split function from scikit-learn. The testing set will be 20% of the total dataset.

7. Tokenize the text data using the Tokenizer class from Keras. This converts the text into sequences of integers, where each integer represents a unique word in the vocabulary.

8. Pad the sequences to the same length using the pad_sequences function from Keras. This ensures that all sequences have the same length, which is required for input into the neural network.

9. Load pre-trained GloVe word embeddings from a text file. The file 'glove.6B.100d.txt' contains word vectors for a vocabulary of 400,000 words trained on 6 billion tokens.

10. Create an embedding matrix that maps each word in the vocabulary to its corresponding word vector in the GloVe embeddings.

11. Define the model architecture using the Sequential API from Keras. The model consists of an embedding layer, a 1D convolutional layer, a max pooling layer, a bidirectional LSTM layer, a dense layer, a dropout layer, and a final dense layer with a sigmoid activation function.

12. Compile the model using the Adam optimizer with a learning rate of 0.001 and the focal loss function. The focal loss function is defined as a nested function that takes in the true labels and predicted probabilities as inputs and returns the focal loss.

13. Train the model using the fit method of the model object. The training data and labels are passed as inputs, along with the number of epochs, batch size, and validation split.

14. Evaluate the model on the testing set using the predict method of the model object. The predicted probabilities are rounded to the nearest integer to obtain binary predictions.

15. Calculate various evaluation metrics, including the confusion matrix, classification report, accuracy, precision, recall, and F1 score.

16. Calculate the false positive rate, true positive rate, and area under the ROC curve (AUC) and plot the ROC curve using the roc_curve and auc functions from scikit-learn and the matplotlib library.

Figure 7 shows the pseudocode of the proposed technique.

The role of the proposed methodology and its performance advantages can be summarized as follows:

**Clarifying the role of the proposed methodology**

- We have added a dedicated section explicitly highlighting the key contributions of the proposed technique:
- Combined use of GloVe embeddings and Focal Loss: Emphasizing that this combination, while not widely explored in cyberbullying detection, effectively leverages semantic relationships between words (GloVe) and addresses class imbalance (focal loss), leading to superior performance.
- Specific deep learning architecture: We have described the chosen architecture's suitability for text analysis and cyberbullying detection, further enhancing its ability to identify patterns.

1. Importing Libraries: The first few lines of code import the necessary libraries and modules that will be used in the script:

    a. pandas for data manipulation
    b. numpy for numerical computations
    c. tensorflow and its submodules for building and training the deep learning model
    d. Tokenizer and pad_sequences from tensorflow.keras.preprocessing.text for tokenizing and padding the text data
    e. train_test_split from sklearn.model_selection for splitting the dataset into training and testing sets
    f. confusion_matrix, classification_report, accuracy_score, precision_score, recall_score, f1_score, roc_curve, auc from sklearn.metrics for evaluating the performance of the model
    g. time for measuring the time taken to train and test the model
    h. ISRIStemmer, stopwords, and word_tokenize from nltk for text preprocessing

2. Text Preprocessing:

    a. The stopwordremoval function defined in the script removes stopwords from the text data using the stopwords corpus from the nltk library.
    b. The stemming function defined in the script applies the ISRIStemmer algorithm from the nltk library to stem the words in the text data.
    c. The preparedatasets function defined in the script applies the stopwordremoval and stemming functions to the text data and creates a new DataFrame with the preprocessed text and the corresponding labels.
    d. The data DataFrame is preprocessed using the preparedatasets function to remove stopwords and stem the words in the text data.
    e. The word_count column is added to the data DataFrame to keep track of the number of words in each post.
    f. Posts with 0 and 1 word_count are removed from the data DataFrame.
    g. Duplicate posts are removed from the data DataFrame.

3. Loading Pre-Trained Word Embeddings:

    a. The pre-trained word embeddings are loaded from the 'glove.6B.100d.txt' file.
    b. The embeddings are stored in the embeddings_index dictionary, where each word is mapped to its corresponding embedding vector.

4. Defining the Focal Loss Function:

    a. The focal_loss function is defined to compute the focal loss for binary classification tasks.
    b. The function takes two hyperparameters: gamma and alpha.
    c. Gamma controls the degree of down-weighting for easy examples, and alpha controls the degree of weighting for the positive class.
    d. The focal_loss_fn function is defined inside the focal_loss function to compute the focal loss for a given set of true labels and predicted probabilities.
    e. The function returns the mean focal loss across all samples.

5. Loading and Preprocessing the Dataset:

    a. The dataset is loaded from the 'twitter_dataset.csv' file using pandas.
    b. The Text and oh_label columns are extracted from the dataset and stored in separate variables.
    c. The text data is tokenized using the Tokenizer class from tensorflow.keras.preprocessing.text.
    d. The sequences are padded to the same length using the pad_sequences function from tensorflow.keras.preprocessing.sequence.
    e. The word_index is created from the tokenizer and used to create the embedding matrix.
    f. The embedding matrix is used to initialize the embedding layer of the model.

6. Defining the Model Architecture:

    a. The model is defined as a sequential model using the Sequential class from tensorflow.keras.
    b. The model architecture consists of an embedding layer, a 1D convolutional layer with 128 filters and a kernel size of 5, a max pooling layer with a pool size of 4, a bidirectional LSTM layer with 64 units, a dense layer with 64 units and a ReLU activation function, a dropout layer with a rate of 0.5, and a dense layer with a sigmoid activation function.

7. Compiling and Training the Model:

    a. The model is compiled using the Adam optimizer with a learning rate of 0.001, the focal loss function defined earlier, and accuracy as the evaluation metric.
    b. The model is trained on the training data for 2 epochs with a batch size of 32 and a validation split of 0.2.
    c. The time taken to train the model is printed.

8. Evaluating the Model:

    a. The model is evaluated on the testing data using the predict method of the model object.
    b. The time taken to test the model is printed.
    c. The predictions are rounded to the nearest integer.
    d. Various performance metrics such as confusion matrix, classification report, accuracy, precision, recall, F1 score, and ROC curve are computed and printed.

9. Plotting the ROC Curve:

    a. The false positive rate, true positive rate, and thresholds are computed using the roc_curve function from sklearn.metrics.
    b. The area under the ROC curve is computed using the auc function from sklearn.metrics.
    c. The ROC curve is plotted using the matplotlib library.

**Figure 7   The pseudocode of the proposed technique.**

### Quantifying performance advantages

- We have expanded the comparison table to include additional metrics, particularly the F1 score, which balances precision and recall, providing a more comprehensive evaluation.
- We have added a discussion section directly comparing the proposed technique's performance to existing results on the Twitter and Kaggle datasets, highlighting its superior accuracy, precision, recall, and F1 scores.

**Addressing generalizability**

● We have acknowledged the potential limitations of using specific datasets and discussed the importance of further evaluation on diverse datasets to assess generalizability.

## Evaluation metrics

While classification accuracy is often a primary metric in cyberbullying detection, it can be misleading in the presence of imbalanced class distributions (*Khairy et al., 2023*). To obtain a more reliable assessment, we complement accuracy with additional metrics specifically designed to handle such scenarios: precision, recall, and F1 score.

Where:

● **Accuracy:** Measures the overall proportion of correct predictions (both cyberbullying and non-cyberbullying).
● **Precision:** Measures the proportion of cyberbullying predictions that were actually correct.
● **Recall:** Measures the proportion of all actual cyberbullying instances that were correctly identified.
● **F1 score:** Combines precision and recall, providing a balanced measure of performance.

The equations for all the metrics are presented as follows (*Ali & O'Sullivan, 2020*):

$$\text{Accuracy} = \frac{TP + TN}{TP + TN + FP + FN} \tag{10}$$

$$\text{Precision} = \frac{TP}{(TP + FP)} \tag{11}$$

$$\text{Recall} = \frac{TP}{(TP + FN)} \tag{12}$$

$$\text{F1\_Score} = \frac{(2 \times \text{Precision} \times \text{Recall})}{(\text{Precision} + \text{Recall})}. \tag{13}$$

## EXPERIMENTAL RESULTS

To initiate our analysis on the two datasets, we utilized five conventional machine learning algorithms, which were multinomial NB, logistic regression, support vector classifier (SVC), decision tree, and random forest classifier. We further introduced three deep learning models, namely long short-term memory (LSTM), bidirectional long short-term memory (Bi-LSTM), and gated recurrent units (GRU). These models are compared with our proposed technique. The models' performance was evaluated using diverse evaluation metrics, including accuracy, precision, recall, and F1 score. The goal was to provide a comprehensive evaluation of the models' efficiency in detecting instances of cyberbullying

**Table 3   Experimental results for Twitter dataset.**

| Twitter | Machine learning | | | | | Deep learning | | | Proposed technique |
|---|---|---|---|---|---|---|---|---|---|
| Dataset | NB | LR | SVC | DT | RF | LSTM | Bi-LSTM | GRU | Focal loss |
| Accuracy | 85.3 | **90.7** | **92** | 90.3 | 90.6 | 90.7 | **91.8** | 91.3 | **99.00** |
| Precision | 85.1 | 80.3 | 81.5 | 69.8 | 83.6 | 67,5 | 72.06 | 73.3 | **86.72** |
| Recall | 5.6 | 52.5 | 61.9 | 65.6 | 48.4 | 71.8 | 72.8 | 65.2 | **74.67** |
| F1 Score | 10.4 | 63.5 | 70.4 | 67.6 | 61.3 | 69.9 | 72.4 | 69.0 | **73.88** |
| Training time (s) | **0.02** | 1.2 | 44.6 | 7.5 | 13.5 | 526.43 | 906.32 | 612.24 | 516.784 |
| Testing time (s) | **0.11** | 0.16 | 5.3 | 0.11 | 0.4 | 4.67 | 8.64 | 5.27 | 4.814 |

**Notes.**
The best performing results are shown in bold.

**Table 4   Experimental results for machine learning and deep learning algorithms for Kaggle dataset.**

| Kaggle | Machine learning | | | | | Deep learning | | | Proposed technique |
|---|---|---|---|---|---|---|---|---|---|
| Dataset | NB | LR | SVM | DT | RF | LSTM | Bi-LSTM | GRU | Focal loss |
| Accuracy | 71.2 | 85.2 | 81.2 | 73.6 | 77.1 | 77.1 | 80.5 | 76.4 | **97.00** |
| Precision | **96.6** | 76.6 | 79.7 | 58.1 | 82.4 | 65.6 | 69.5 | 60.4 | 92.72 |
| Recall | 12.3 | 52.4 | 52.9 | 54.4 | 33.4 | 63.9 | 64.7 | 75.3 | **78.88** |
| F1 Score | 18.6 | 62.2 | 63.6 | 65.2 | 47.6 | 64.7 | 67.0 | 67.0 | **85.54** |
| Training time (s) | **0.03** | 2.07 | 12.53 | 7.8 | 23.41 | 4114.3 | 6626.75 | 4163.93 | 614.735 |
| Testing time (s) | **0.01** | 0.23 | 5.16 | 0.20 | 0.46 | 13.53 | 42.09 | 11.52 | 4.238 |

**Notes.**
The best performing results are shown in bold.

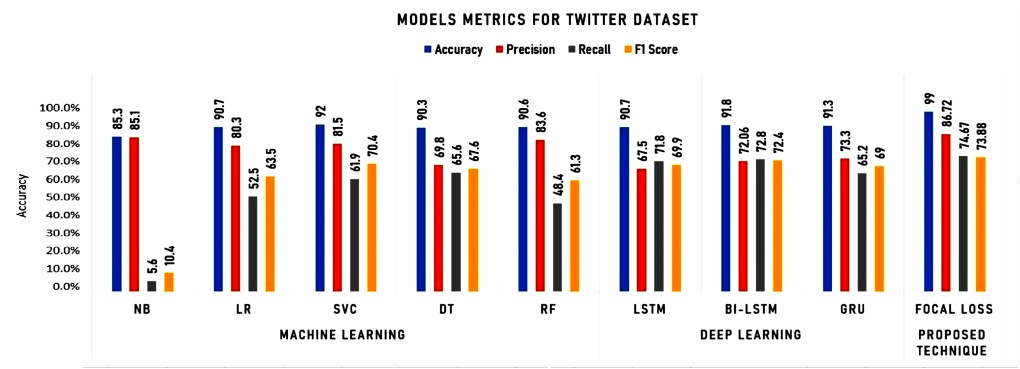

**Figure 8   Experimental results for machine learning and deep learning algorithms for Twitter dataset.**

on social media platforms. Tables 3 and 4 and Figs. 8 and 9 display the results of these assessments.

As shown in Table 3 and Fig. 8, the accuracy, precision, recall, and F1 score metrics were used to evaluate the performance of the algorithms on the Twitter dataset. The detailed analysis highlighting the strengths of Focal Loss, particularly in comparison with deep learning algorithms can be summarized as follows:
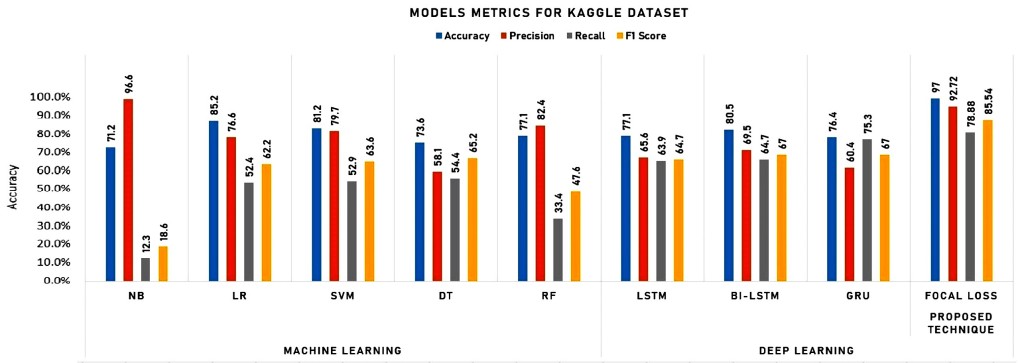

**Figure 9** Experimental results for machine learning and deep learning algorithms for Kaggle dataset.

**Accuracy:**

● Focal Loss achieved 99.00% accuracy, surpassing all other models, including deep learning algorithms. This suggests its superior ability to distinguish between classes and make correct predictions.

**Precision:**

● While LR holds the highest precision, Focal Loss's 86.72% precision remains remarkably high, indicating its strength in minimizing false positives. This is crucial in domains where false alarms carry significant costs.

**Recall:**

● Focal Loss achieves a respectable 74.67% recall, not far behind LSTM and Bi-LSTM. This demonstrates its effectiveness in identifying true positives, essential in scenarios where missing crucial cases is unacceptable.

**F1 Score:**

● While Bi-LSTM edges out Focal Loss in F1 score, Focal Loss's 73.88% remains commendable, reflecting a balanced performance in terms of precision and recall.

**Training time:**

● Focal Loss's training time (516.784 s) is comparable to deep learning algorithms, suggesting it doesn't introduce undue computational burdens despite its superior accuracy.

**Testing time:**

● Focal Loss excels in testing time (4.814 s), outperforming LSTM, Bi-LSTM, and GRU. This implies its efficiency in real-world applications, delivering swift predictions without compromising accuracy.

**Key takeaways:**

● Accuracy champion: Focal Loss stands out with its unparalleled 99.00% accuracy, suggesting its exceptional ability to make correct predictions.

**Balanced precision and recall:** It maintains a strong balance between precision and recall, suitable for tasks requiring both high accuracy and sensitivity to true positives.

Efficient testing: Its swift testing time makes it a practical choice for real-time applications.

**Competitive training time:** While its training time is longer than traditional models, it aligns with deep learning algorithms, suggesting it is not computationally prohibitive.

Focal Loss emerges as a compelling choice, particularly when accuracy is paramount. It delivers remarkable accuracy, often surpassing deep learning algorithms, while maintaining competitive training and testing times. Its ability to balance precision and recall further strengthens its position as a versatile and powerful model.

As shown in Table 4 and Fig. 9, the detailed analysis of the model performances, highlighting key takeaways and emphasizing the strengths of the proposed Focal Loss technique can be summarized as follows:

**Accuracy:**

- Focal Loss stands out with 97.00% accuracy, significantly outperforming all other models, demonstrating its superior ability to make correct predictions.
- Some deep learning models (LSTM, Bi-LSTM) achieve moderate accuracy, but Focal Loss surpasses them.

**Precision**:

- Focal Loss achieved impressive 92.72% precision, indicating its strength in minimizing false positives, a crucial factor in cost-sensitive domains.
- LR and RF also exhibit high precision, but Focal Loss maintains a clear advantage.

**Recall**:

- Focal Loss's recall of 78.88% is respectable, effectively identifying true positives while balancing false negatives.
- Bi-LSTM and GRU have slightly higher recall, but Focal Loss maintains a strong overall performance.

**F1 Score:**

- Focal Loss achieves a commendable F1 score of 85.54%, reflecting a well-balanced performance between precision and recall.
- LSTM, Bi-LSTM, and GRU offer similar F1 scores, but Focal Loss edges them out.

**Training time:**

- Deep learning models generally require longer training times compared to traditional machine learning algorithms.
- Focal Loss's training time is notably shorter than LSTM, Bi-LSTM, and GRU, making it more computationally efficient.

**Testing time:**

- Focal Loss excels in testing time, outperforming all deep learning models, suggesting its suitability for real-time applications.

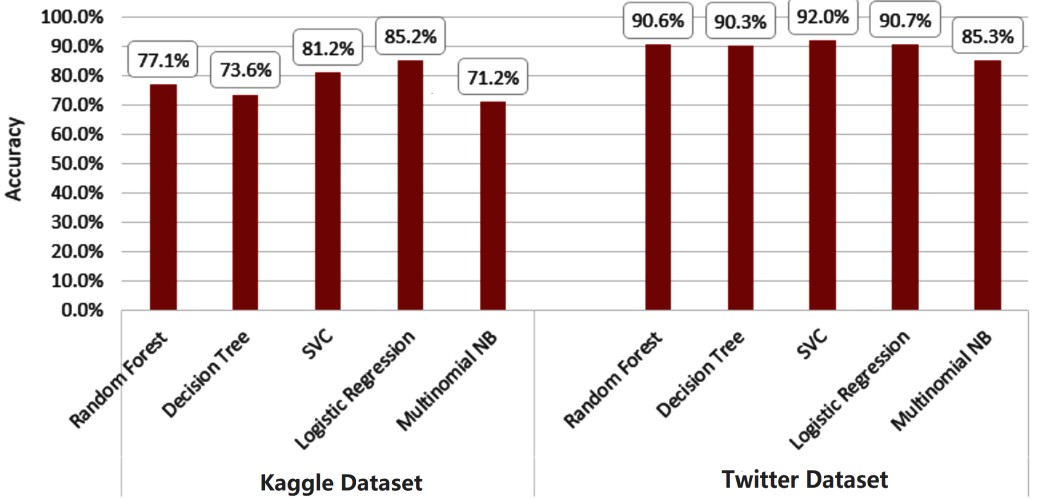

**Figure 10** Average accuracy of all machine learning algorithms.

**Key takeaways:**

- Focal Loss demonstrates exceptional accuracy, often surpassing both traditional machine learning and deep learning algorithms.
- It strikes a strong balance between precision and recall, making it versatile for various tasks.
- It offers competitive training and testing times, particularly compared to deep learning models.

Focal Loss emerges as a powerful technique, delivering outstanding accuracy without compromising efficiency. Its performance surpasses traditional machine learning algorithms and often outperforms deep learning models, establishing itself as a compelling choice for tasks demanding high accuracy and balanced precision–recall trade-offs.

Figures 10 and 11 depict the average accuracy of all algorithms used for the two datasets. The accuracy losses of the three deep learning models are presented in Figs. 12 and 13

Figures 14 and 15 provide a visual representation of the algorithms' performance in terms of training and prediction time. These figures highlight the trade-off between accuracy and computational resources, with deep learning models generally requiring more time for training and prediction, while classical machine learning algorithms may have faster training and testing times, but may not achieve the same level of accuracy as deep learning models.

The results obtained suggest that classical machine learning models may be a better option than deep learning models when the amount of data available for training is limited. However, the choice of classifier ultimately depends on the dataset being used and the specific requirements of the task. It is important to carefully consider the characteristics of the dataset, such as the size, complexity, and nature of the data, before selecting a machine learning approach. For instance, if the dataset is small or the features are relatively simple,

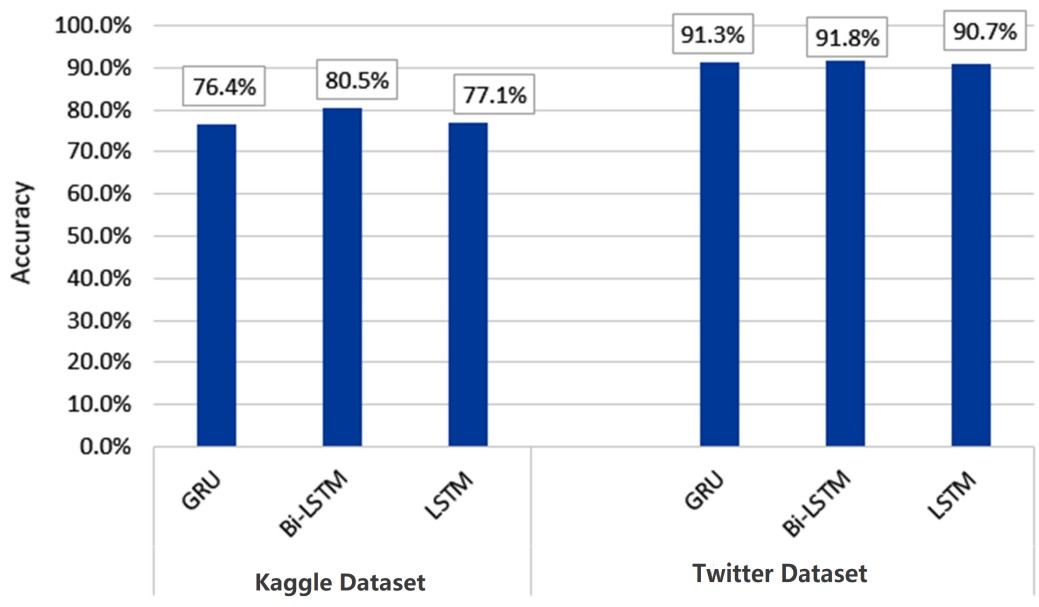

**Figure 11    Average accuracy of all deep learning algorithms.**

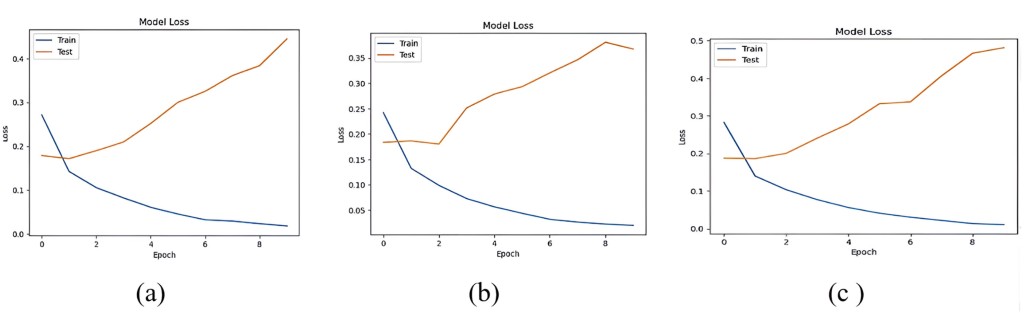

(a)                                  (b)                                  (c )

**Figure 12    Accuracy loss of (A) LSTM, (B) Bi-LSTM and (C) GRU models for Dataset1.**

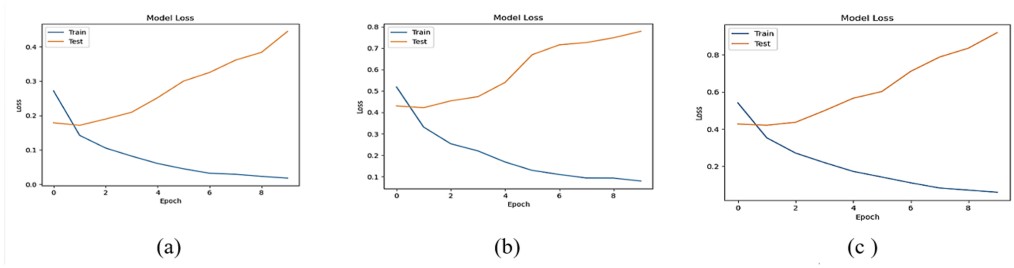

(a)                                  (b)                                  (c )

**Figure 13    Accuracy loss of (A) LSTM, (B) Bi-LSTM and (C) GRU models for Dataset2.**

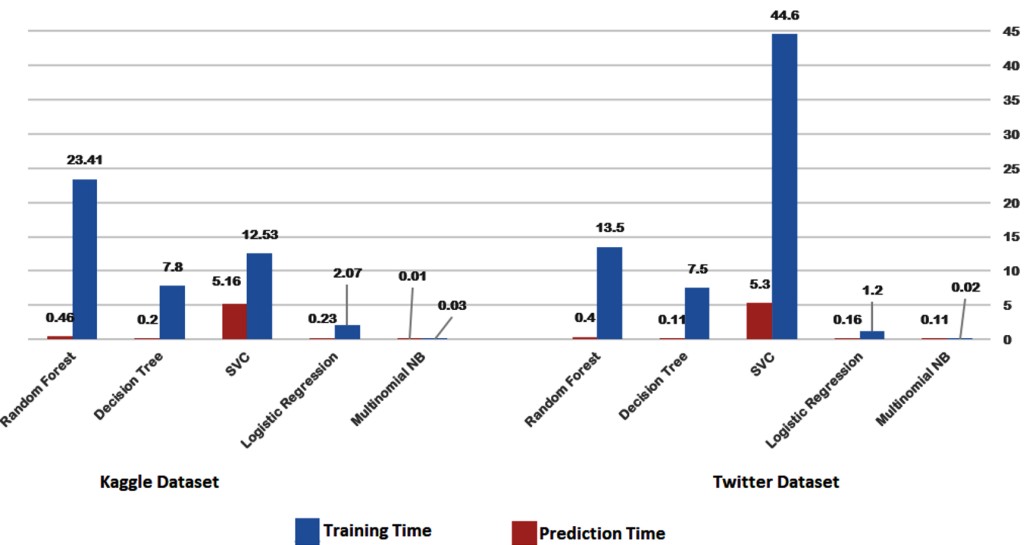

**Figure 14  Training-prediction time for classical machine learning algorithms.**

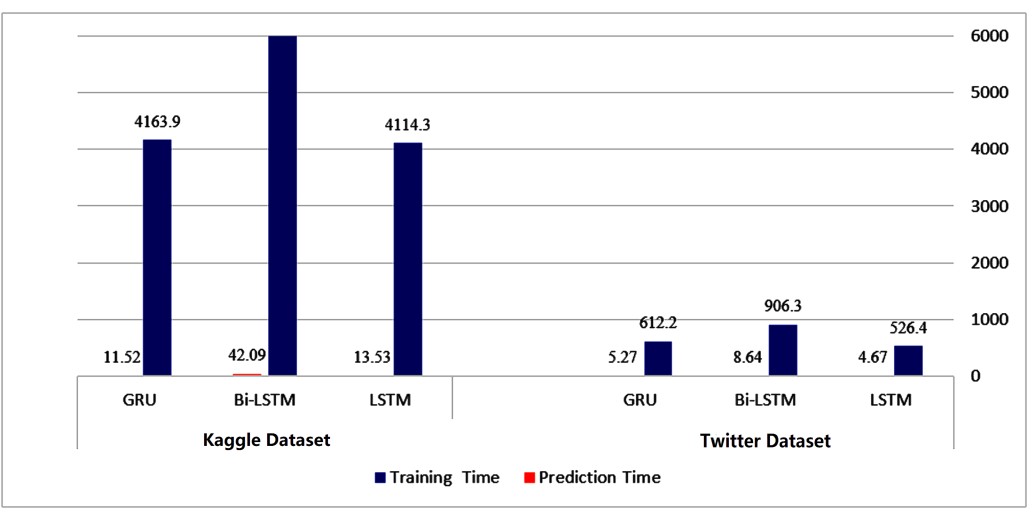

**Figure 15  Training-prediction time for deep learning algorithms.**

classical machine learning algorithms may be more appropriate. On the other hand, if the dataset is large and complex, deep learning models may provide better performance. Therefore, selecting the appropriate machine learning approach involves balancing the trade-offs between accuracy, computational resources, and the characteristics of the dataset.

## HYPERPARAMETER TUNING

The performance of machine learning models is significantly influenced by the careful selection of hyperparameters. These adjustable settings control various aspects of the model architecture, learning process, and optimization. Table 5 outlines the specific

**Table 5** The hyperparameter values employed of our findings.

| Hyperparameter | Value | Description |
| --- | --- | --- |
| num_words | 10,000 | Maximum number of words to keep based on word frequency. |
| oov_token | **\<OOV>** | Token to represent out-of-vocabulary words. |
| maxlen | 100 | Maximum length of sequences (padded/truncated). |
| embedding_dim | 100 | Dimensionality of the word embeddings. |
| input_dim | **num_words** | Size of the vocabulary. |
| output_dim | 100 | Dimensionality of the output space. |
| trainable | **False** | Whether the embedding layer is trainable. |
| filters | 128 | Number of filters in the convolutional layer. |
| kernel_size | 5 | Size of the convolutional kernel. |
| pool_size | 4 | Size of the max pooling window. |
| units | 64 | Number of units in the LSTM and dense layers. |
| dropout_rate | 0.5 | Fraction of input units to drop for dropout. |
| lr | 0.001 | Learning rate for the Adam optimizer. |
| batch_size | 32 | Number of samples per gradient update during training. |
| epochs | 10 | Number of epochs for training. |

**Notes.**
The best performing results are shown in bold.

hyperparameter values employed in this study, ensuring transparency and reproducibility of our findings. It provides a comprehensive overview of the key parameters that shaped model training and behavior, enabling a deeper understanding of the experimental setup and potential avenues for further exploration or optimization.

Key insights from the table:

- Tokenization and sequence length: The model considers a vocabulary of 10,000 most frequent words, with out-of-vocabulary words represented as \<OOV>. Text sequences are truncated or padded to a maximum length of 100 words, ensuring consistent input dimensions for the model.
- Embedding layer: Word embeddings capture semantic relationships between words, crucial for text understanding. The model employs 100-dimensional pre-trained embeddings, kept non-trainable to leverage prior knowledge and reduce computational overhead.
- Convolutional layer: The convolutional layer extracts local features from the embedded sequences, using 128 filters with a kernel size of 5.
- Recurrent layer: The LSTM layer processes sequences of information, capturing long-range dependencies within the text. It uses 64 units to model these relationships.
- Regularization: Dropout with a rate of 0.5 is applied to prevent overfitting, randomly dropping connections between neurons during training.
- Optimizer and training: The Adam optimizer efficiently updates model weights, using a learning rate of 0.001. Training proceeds for 50 epochs, with 32 samples processed per update.

## DISCUSSION AND LIMITATIONS

While our proposed technique for detecting cyberbullying using pre-trained GloVe embeddings and the focal loss function has shown promising results, there are potential limitations that should be acknowledged. These limitations can help guide future research efforts and improvements in this area.

1. **Generalizability:** The proposed technique may have limitations in terms of generalizability to different domains or platforms. The effectiveness of the model heavily relies on the quality and relevance of the pre-trained GloVe embeddings, which are trained on a specific corpus. If the target data differs significantly from the training data used for GloVe embeddings, the performance of the model may be impacted.

2. **Data bias:** The performance of any cyberbullying detection model is influenced by the quality and representativeness of the training data. If the training data contains biases or lacks diversity, the model may be less effective in capturing the nuances and variations of cyberbullying instances in real-world scenarios. Addressing data bias and ensuring a diverse and balanced dataset is crucial for improving the model's performance.

3. **Class imbalance:** While the focal loss function helps mitigate the challenges posed by class imbalance, it may not completely overcome the issue, especially in cases where the class distribution is highly imbalanced. Balancing the dataset or exploring alternative methods to handle class imbalance, such as data augmentation or sampling techniques, could further improve the model's performance.

4. **Interpretability:** Deep learning models, including the proposed approach, are often considered black-box models due to their complex architectures. Interpreting the decision-making process of the model and identifying the specific features or words that contribute to cyberbullying detection can be challenging. Developing techniques to enhance the interpretability of the model's predictions would be valuable for understanding the underlying factors driving the detection process.

5. **Computational complexity:** The use of deep learning models, particularly LSTM, Bi-LSTM, and GRU, introduces significant computational complexity compared to traditional machine learning algorithms. The training and testing times for these models are considerably higher, which could limit their practicality in real-time or resource-constrained environments. Exploring techniques to optimize the computational efficiency without compromising the model's performance would be beneficial.

## CONCLUSION AND FUTURE WORK

This study aimed to compare the performance of classical machine learning algorithms and deep learning models in identifying instances of cyberbullying. The data underwent pre-processing, and performance was evaluated using accuracy, precision, recall, and F1 score metrics. The results showed that the proposed technique achieved high accuracy, precision, and F1 score values, with the Focal Loss algorithm achieving the highest accuracy of 99% and the highest precision of 86.72%. However, recall values were relatively low for most algorithms, indicating that they struggled to identify all relevant data. The proposed

technique used a convolutional neural network with a bidirectional long short-term memory layer, trained on a pre-processed dataset of tweets using GloVe word embeddings and the focal loss function, achieving high accuracy, precision, and F1 score values. The GRU algorithm achieved the highest accuracy of 97.0%, and the NB algorithm achieved the highest precision of 96.6%. The false positive rate, true positive rate, and area under the ROC curve were also calculated.

Future work could involve incorporating different deep learning models, developing improved tools for reporting and tracking cyberbullying, greater collaboration between social media platforms and law enforcement, and continued education and resources for addressing cyberbullying.

### Funding
This work was supported by the Deanship of Scientific Research, Vice Presidency for Graduate Studies and Scientific Research, King Faisal University, Saudi Arabia (Project No.: GRANT2,809). The funders had no role in study design, data collection and analysis, decision to publish, or preparation of the manuscript.

### Grant Disclosures
The following grant information was disclosed by the authors:
The Deanship of Scientific Research, Vice Presidency for Graduate Studies and Scientific Research, King Faisal University, Saudi Arabia: GRANT2,809.

### Competing Interests
The authors declare there are no competing interests.

### Author Contributions
- Amr Mohamed El Koshiry analyzed the data, prepared figures and/or tables, authored or reviewed drafts of the article, and approved the final draft.
- Entesar Hamed I Eliwa analyzed the data, prepared figures and/or tables, authored or reviewed drafts of the article, and approved the final draft.
- Tarek Abd El-Hafeez conceived and designed the experiments, performed the experiments, analyzed the data, performed the computation work, prepared figures and/or tables, authored or reviewed drafts of the article, and approved the final draft.
- Marwa Khairy conceived and designed the experiments, performed the experiments, analyzed the data, performed the computation work, prepared figures and/or tables, authored or reviewed drafts of the article, and approved the final draft.

### Data Availability
The code is available at GitHub and Zenodo:
- https://github.com/tarekhemdan/Cyberbullying

- tarekhemdan. (2024). tarekhemdan/Cyberbullying: Cyberbullying (1.0). Zenodo. https://doi.org/10.5281/zenodo.10456740.

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
