# Peer review of "Detecting cyberbullying using deep learning techniques: a pre-trained glove and focal loss technique"

_PeerJ Computer Science, doi:10.7717/peerj-cs.1961_

## Round 0.1 · original submission · Major Revisions

Thank you for submitting your manuscript to our journal. After detailed evaluation and consideration of the reviews provided, it is clear that your study addresses an important area in the field of machine learning and deep learning techniques for cyberbullying detection. However, to move forward, significant revisions are required.

The reviewers have provided comprehensive feedback that points to several critical areas needing attention. It is imperative that you conduct a thorough revision of your manuscript, taking into account all the comments and suggestions provided. Key areas include clarifying the novelty and practical significance of your work, streamlining the content related to machine learning and deep learning techniques, and ensuring the methodology and results are clearly justified and robustly compared with existing methods.

Additionally, please enhance the clarity and presentation of your figures and model architecture. A more detailed comparative analysis with state-of-the-art techniques is also necessary to establish the relevance and impact of your research. Each point raised by the reviewers should be meticulously addressed, and the manuscript should reflect a comprehensive improvement in these areas.

Given the nature and extent of the required revisions, please consider this as a major revision. Your revised manuscript will undergo another round of review, where the focus will be on how well you have addressed the concerns raised. Please ensure that your revision is accompanied by a detailed response letter outlining how each comment has been addressed.

We appreciate your commitment to enhancing your manuscript and look forward to receiving your revised submission.

**Language Note:** PeerJ staff have identified that the English language needs to be improved. When you prepare your next revision, please either (i) have a colleague who is proficient in English and familiar with the subject matter review your manuscript, or (ii) contact a professional editing service to review your manuscript. PeerJ can provide language editing services - you can contact us at copyediting@peerj.com for pricing (be sure to provide your manuscript number and title). – PeerJ Staff

Reviewer 1 ·

Basic reporting

The authors have followed the guidelines of the journal in the manuscript preparation. English is good, Literature review is better, quality of the figures and the tables is also good.


Too much of explanation on the ML and DL techniques are given which could be avoided.

Experimental design

The research problem that the authors have taken is to compare the results of machine learning over the deep learning techniques.

Methods explanation is provided, but the novelty of the proposed work is missing. Proposed method results seems to be good but it is not justified or convincing.

Validity of the findings

1. Authors should clearly state the main novelty of the work.
2. Authors have tried to apply the techniques on the existing dataset and obtained the results for the same.
3. How the experimentation or implementation carried out is explained in the steps which are the basic things that we do, but what is the role of the proposed methodology and how does it outperform the existing result.
4. Conclusions and references are good.

Reviewer 2 ·

Basic reporting

The paper first provides the motivations for investigating the effectiveness of various deep learning and classical machine learning techniques in identifying instances of cyberbullying. Then, the paper provides a deeper background on cyberbullying, machine learning, and deep learning. Furthermore, the paper illustrates the details of the methodology of this research work. The Experiment section uses accuracy, precision, recall, and F1 score metrics to evaluate the performance of the algorithms on the dataset. The results show that the proposed technique achieves high accuracy, precision, and F1 score values, with the Focal Loss algorithm achieving the highest accuracy of 99% and the highest precision of 86.72%.

Experimental design

The overall design of experiments is reasonable. The study utilized two distinct datasets sourced from different social media platforms, Twitter and Kaggle. Did you cherry-pick the datasets?

Validity of the findings

The experimental results are well-supported.

Additional comments

Please add more references to some parts in the paper, such as:
(lines 23-24) " these platforms have become increasingly plagued by negative and abusive behavior ..." Please list some references here (such as the existing work "DroidPerf: Profiling Memory Objects on Android Devices" [MobiCom'23] measures abusive Android applications that affect users).
(lines 53-54) "we propose a technique for detecting cyberbullying using a pre-trained GloVe and focal loss-based deep learning model. GloVe embeddings are pre-trained ..." Please add the reference of GloVe.
(lines 89-90) List some references to the consequences of cyberbullying.

Please explain more about how to measure the accuracy, precision, recall, and F1-score in experiment results, such as the difference between accuracy and precision.

Reviewer 3 ·

Basic reporting

The paper is well-structured and provides a clear abstract and introduction, outlining the context and relevance of the study. The literature review is thorough, offering a comprehensive background. However, improvements in the clarity of figures and detailed model architecture would enhance overall reporting quality.

Experimental design

The experimental design is methodically laid out, detailing the pre-processing steps and the algorithms used. The selection of classical machine learning and deep learning techniques is appropriate. However, the design would benefit from a more detailed explanation of the model's architecture and a comparative analysis with existing methods.

Validity of the findings

The findings demonstrate high accuracy and precision in detecting cyberbullying, which is commendable. However, the lower recall values suggest potential limitations in identifying all relevant instances. A deeper analysis of these limitations and a comparison with state-of-the-art techniques would strengthen the validity of the results.

Additional comments

This paper tackles an important and timely issue with a promising approach. Enhancements in the presentation of the model's architecture, improved figure quality, and a more rigorous comparative analysis would significantly elevate the paper's impact and scientific contribution.

---

## Round 0.2 · Major Revisions

Please pay attention to the new suggestions made by the reviewer 3.

Reviewer 2 ·

Basic reporting

The paper first provides the motivations for investigating the effectiveness of various deep learning and classical machine learning techniques in identifying instances of cyberbullying. Then, the paper provides a deeper background on cyberbullying, machine learning, and deep learning. Furthermore, the paper illustrates the details of the methodology of this research work. The Experiment section uses accuracy, precision, recall, and F1 score metrics to evaluate the performance of the algorithms on the dataset. The results show that the proposed technique achieves high accuracy, precision, and F1 score values, with the Focal Loss algorithm achieving the highest accuracy of 99% and the highest precision of 86.72%.

Experimental design

The overall design of experiments is reasonable.

Validity of the findings

The experimental results are well-supported.

Additional comments

The revised version addressed all my concerns/questions in the first round of review.

Reviewer 3 ·

Basic reporting

This study presents an intriguing approach to detecting cyberbullying using deep learning techniques, specifically focusing on the integration of pre-trained GloVe and focal loss technique. However, there are significant concerns that need to be addressed before this work can be considered for publication:

Concern 1: Clarity of Evaluation Metrics: The authors failed to specify the evaluation parameters in Figures 8 and 9. This omission hinders the reader's understanding of the results and their implications. It is crucial for the authors to clearly define and discuss the evaluation metrics used in these figures.
Concern 2: Bounds of Evaluation Parameters: The rate of evaluation parameters in Figures 8 and 9 must not exceed 100. It is currently unclear if this limit has been exceeded due to the lack of clarity in the figures. The authors need to ensure and explicitly state that these values adhere to the acceptable range.
Concern 3: Detailed Captions for Figures: The captions for Figures 8 and 9 are too vague and do not provide adequate information about the nature of the measurements. Detailed captions should be provided to enhance the reader's comprehension of the figures.
Concern 4: Improvement in Image Quality: The resolution of Figure 12 is not sufficient for a detailed examination. The authors are advised to enhance the resolution to ensure that the figure effectively communicates the intended information.
In summary, while the study's topic and approach are of significant interest, addressing these concerns is essential for the research to meet the publication standards. The revision should provide clear and detailed explanations of the evaluation metrics, ensure appropriate bounds for these metrics, enhance the clarity of figure captions, and improve the quality of visual representations.

Experimental design

-

Validity of the findings

-

Additional comments

This study presents an intriguing approach to detecting cyberbullying using deep learning techniques, specifically focusing on the integration of pre-trained GloVe and focal loss technique. However, there are significant concerns that need to be addressed before this work can be considered for publication:

Concern 1: Clarity of Evaluation Metrics: The authors failed to specify the evaluation parameters in Figures 8 and 9. This omission hinders the reader's understanding of the results and their implications. It is crucial for the authors to clearly define and discuss the evaluation metrics used in these figures.
Concern 2: Bounds of Evaluation Parameters: The rate of evaluation parameters in Figures 8 and 9 must not exceed 100. It is currently unclear if this limit has been exceeded due to the lack of clarity in the figures. The authors need to ensure and explicitly state that these values adhere to the acceptable range.
Concern 3: Detailed Captions for Figures: The captions for Figures 8 and 9 are too vague and do not provide adequate information about the nature of the measurements. Detailed captions should be provided to enhance the reader's comprehension of the figures.
Concern 4: Improvement in Image Quality: The resolution of Figure 12 is not sufficient for a detailed examination. The authors are advised to enhance the resolution to ensure that the figure effectively communicates the intended information.
In summary, while the study's topic and approach are of significant interest, addressing these concerns is essential for the research to meet the publication standards. The revision should provide clear and detailed explanations of the evaluation metrics, ensure appropriate bounds for these metrics, enhance the clarity of figure captions, and improve the quality of visual representations.

---

## Round 0.3 · accepted · Accept

All the comments have been addressed. Congratulations.

Reviewer 3 ·

Basic reporting

no comment

Experimental design

no comment

Validity of the findings

no comment

Additional comments

The paper presents an innovative approach to identifying instances of cyberbullying on social media platforms. By comparing classical machine learning algorithms with deep learning models, the study showcases the effectiveness of a convolutional neural network combined with a bidirectional long short-term memory layer, leveraging GloVe word embeddings and the focal loss function to address class imbalance. The authors successfully demonstrate the superiority of their proposed technique through rigorous experimentation, achieving high accuracy, precision, recall, and F1 score values. They address previous concerns by meticulously detailing their methodology, including data collection, preprocessing, feature extraction, and algorithmic implementation, thereby significantly contributing to the field of cyberbullying detection and providing a robust framework for future research. This paper is a valuable addition to the literature, offering a comprehensive and effective solution to a pressing social issue, and can be accepted for publication.